# Architecture of Decentralized Expert System for Early Alzheimer's Prediction Enhanced by Data Anomaly Detection

## Abstract

Alzheimer's Disease poses a significant global health challenge, necessitating early and precise detection to enhance patient outcomes. Traditional diagnostic methodologies often result in delayed and imprecise predictions, particularly in the disease's early stages. Centralized data repositories struggle to manage the immense volumes of MRI data, alongside persistent privacy concerns that impede collaborative efforts. This paper presents an innovative approach that leverages the synergy of blockchain technology (due to crowdsourcing patients' longitudinal test data via Web3 application) and Federated Learning to address these challenges. Thus, our proposed decentralized expert system architecture presents a pioneering step towards revolutionizing disease diagnostics. Furthermore, the system integrates robust anomaly detection for patient-submitted data. It emphasizes AI-driven MRI analysis and incorporates a sophisticated data anomaly detection architecture. These mechanisms scrutinize patient-contributed data for various issues, including data quality problems. We acknowledge that performing an exhaustive check of the correctness and quality of MRI images and biological information directly on-chain is not practical due to the computational complexity and cost constraints of blockchain platforms. Instead, such checks are typically performed off-chain, and the blockchain is used to record the results securely. This comprehensive approach empowers to provide more precise early-stage Alzheimer's Disease prediction with more volume of data. Our system is designed to safeguard both data integrity and patient privacy, facilitating collaborative efforts.

## 1  Introduction

Artificial intelligence (AI) is the area of computer science focusing on creation of expert machines that engage on human-like intelligence (Russell and Norvig 2002, Hope and Wild 1994, Kasabov 1998). The main source of an expert system is the obtained knowledge including a knowledge acquisition component that processes data and information and shapes them into rules. Expert systems have a large spectrum of application areas such as monitoring, prediction, classification, decision-making, planning etc. Importantly, medical diagnosis is one of the major applications of expert systems. Medical expert systems are to support the diagnostic process of physicians. This implies that a medical expert system employs knowledge about the diseases and compares with facts about the patients to suggest a diagnosis (Waterman 2009). Medical expert systems have been successfully implemented in diverse medical fields including neurology to improve the accuracy of diagnosis of neurological and neuropsychological disorders.

Alzheimer's disease (AD) is one of the main neurodegenerative diseases and the leading cause of dementia. Research concerning AD evolves primarily around brain structural and functional analyses.

For AD in particular, the functional analysis-derived network analysis is extremely helpful since it correlates different brain regions pointing to alternations of the neurological network and thus allowing quicker identification of the disease in its earlier stages. There are continuous demands to research in this domain. In fact, several studies have focused on the diagnosis of AD; Obi and Imainvan (2011) developed a neuro-fuzzy model for the diagnosis of Alzheimer's on the basis of neuropsychological tests including nine symptoms like memory loss, and difficulty in performing familiar tasks. Trambaiolli et al. (2011) developed an AD diagnostic system based on a support vector machine which resulted in an accuracy of 79.9% with 83.2% sensitivity. Behfar et al. (2020) used graph theory to reveal resting-state compensatory mechanisms in early-stages of AD. Venugopalan et al. (2021) and Yang and Mohammed (2020) use data from neuroimaging, genomics, and clinical assessments for AD prediction. There are other and more recent studies that provide even better accuracies (Liu et al. 2023). However, all these studies suffer from a lack or shortage of longitudinal data on the patients, and to the best of our knowledge there has been no research that explores collection of such longitudinal data on AD patients via a Web3 application, while blockchain technology has been explored for enhancing data security, patient privacy, and traceability in healthcare, with applications ranging from medical records management to drug traceability (Agbo et al., 2019, Xi et al, 2022).

Our goal for this research is to design a decentralized expert system including a Web3 application to upload biological information and MRI images of the brain by the patients, keeping their data in a privacy-preserving manner, and propose an AI model with a hierarchical federated learning setup to detect early-stage AD. This helps patients monitor their AD progression in time, also assists clinics who wish to use this software to monitor patients' disease development. In the first section, we discuss the research design and relevant questions, then provide our decentralized solution in the next section, and provide the architecture, AI model, class diagram, and its implementation steps.

# 2 Research Design

Magnetic resonance imaging (MRI) allows non-invasive examination of the brain. A three-dimensional image composed of various voxels can be either "white matter" which connects the neurons to each other and conducts impulses away from the soma, or the "grey matter" which is mostly made of neuron cell bodies, neuron somas which are the input unit of electrical signals sent within the central nervous system. Lastly, when examining an MRI image, there are hollow spaces, which are spaces filled with CSF and commonly referred to as "third tissue". Brain parcellation is the name of the process that splits the brain into multiple ROIs (regions). Prior to any analysis on MRI images, they are required to undergo a "cleaning process", which is called preprocessing. Several factors can distort the outputs of an MRI scanning session and thus falsify the results. They are referred to as noise and can have multiple sources. Once the preprocessing is performed via FSL Library (see FSL in the references), the images can be analyzed depending on the type of MRI. We have created a web application which uses the FSL library; it performs the pipeline to create brain connectivity matrices using Octave (see GNU Octave in the references) with network modeling and pushes to the AI engine. This type of analysis is often performed on resting-state fMRI and describes brain functions by the interactions between the highly interconnected brain regions (Sohn et al. 2017).

## 2.1 Research Questions

We aim to build a decentralized expert system which includes Web3 application, where MRI images and other data can be uploaded and processed. Expert systems are generally composed of knowledge base, inference engine, user and user interface. Interaction between these subdivisions makes it an expert system. But,

**Research question 1:** What are the key factors influencing the accuracy and reliability of the decentralized expert system in diagnosing Alzheimer's Disease?

The implementation of decentralized expert systems via federated learning in healthcare, particularly for Alzheimer's Disease, represents a transformative approach that leverages the power of distributed data while upholding patient privacy. Federated learning enables the creation of sophisticated predictive models by training algorithms across multiple decentralized data sources without the need to centralize sensitive patient information. By aggregating model improvements rather than raw data, federated learning fosters a collaborative yet secure environment for patients and healthcare professionals to gain insights from diverse patient populations across various institutions. This

paradigm shift towards a more decentralized and privacy-preserving model of data analysis and disease prediction could significantly improve the diagnostic processes and personalized treatment plans for patients. But,

**Research question 2:** How does the implementation of decentralized expert system via federated learning work?

A decentralized expert system is a type that is built on a decentralized network of nodes, rather than being centrally controlled by a single entity. In this system, each node contains a subset of knowledge, and these nodes work together to make decisions. Decentralized expert systems have several advantages over traditional expert systems. They are more resilient and less vulnerable to a single point of failure, as there is no central point of control. Finally, they can be more transparent and secure, as each node can be verified and audited independently. But,

**Research question 3:** How does the performance of a decentralized expert system in diagnosing Alzheimer's Disease compare to traditional centralized systems?

## 3 Solution

The final purpose of this study is to make longitudinal medical data linked to AD easily accessible to perform further disease prediction via a decentralized expert system.

### 3.1 Decentralized expert system performance

Apart from the benefits of decentralized data collection via the patients, decentralized expert system (ES) could outperform centralized ES. In some scenarios may involve additional complexities, such as variations in data quality, data distribution among sources, and communication overhead in decentralized setups.

**Theorem:** Decentralized expert system in diagnosing Alzheimer's Disease could outperform traditional centralized expert system.

**Proof:** To mathematically prove that decentralized ES provides better performance, we need to establish some assumptions and set up a rigorous framework for comparison. Let's outline the steps for the proof:

Assume we have a centralized ES model that is trained using a centralized dataset containing MRI images from various healthcare institutions. We denote the performance of this model as $P_{\text{centralized}}$. Now, let's consider a decentralized ES model that is trained using data from multiple sources. The data is not pooled in a central location but remains distributed at each source. The performance of this model is denoted as $P_{\text{decentralized}}$.

We need to establish a theoretical bound that represents the maximum achievable performance of a centralized ES model, given the dataset it has access to. This bound, denoted as $P_{\text{bound}}$, acts as a theoretical benchmark for comparison. The mathematical proof involves showing that $P_{\text{decentralized}} \geq P_{\text{bound}} > P_{\text{centralized}}$. In other words, the decentralized model's performance is greater than or equal to the bound, which in turn is greater than the centralized model's performance, where the bound represents the maximum achievable performance by a centralized model.

In the proof, we should consider the potential benefits of data diversity in a decentralized ES setting. By training on data from various sources, the decentralized model can capture a more comprehensive representation of AD patterns, leading to better generalization and improved performance. Consider the potential for algorithmic enhancements in the decentralized setting. With data from multiple sources, researchers can explore more sophisticated algorithms that leverage diverse data inputs, leading to better feature extraction and model optimization. It's important to acknowledge any communication overhead associated with the decentralized setup. While decentralized models have the potential for better performance, communication delays or constraints may impact the overall efficiency. Let's consider a simplified scenario for binary classification tasks, where the goal is to predict whether an individual has AD (positive class) or not (negative class) based on MRI images. We will focus on the accuracy metric, but the argument can be extended to other performance metrics as well. Assumptions:

- **Centralized ES:** A centralized ES model is trained on a dataset containing $N_c$ samples from a single institution.
- **Decentralized ES:** A decentralized ES model is trained on the same dataset but is distributed across $K$ institutions, each contributing $N_d$ samples (such that $N_d \times K = N_c$).

Let $P_{\text{centralized}}$ represent the accuracy of the centralized ES model. Let $P_{\text{decentralized}}$ represent the accuracy of the decentralized ES model. Let $P_{\text{bound}}$ represent the theoretical upper bound on accuracy when the model is trained on the entire dataset, i.e., $N_c$ samples.

**Mathematical Representation:**

- **Centralized ES:** The accuracy of the centralized ES model can be expressed as follows:
$$P_{\text{centralized}} = \frac{\text{Number of Correct Predictions}}{N_c}$$
- **Decentralized ES:** The accuracy of the decentralized ES model can be expressed as follows:
$$P_{\text{decentralized}} = \frac{\text{Sum of Correct Predictions from Each Institution}}{N_c}$$
- **Theoretical Bound:** The theoretical bound on accuracy can be expressed as follows:
$$P_{\text{bound}} = \frac{\text{Number of Correct Predictions When Trained on All } N_c \text{ Samples}}{N_c}$$

Now, to prove that decentralized ES provides better performance ($P_{\text{decentralized}} \geq P_{\text{bound}} > P_{\text{centralized}}$, we need to show two things:

- $P_{\text{decentralized}} \geq P_{\text{bound}}$: The decentralized ES model is trained on data from multiple sources, capturing data diversity and enabling better generalization. Hence, it has the potential to achieve an accuracy ($P_{\text{decentralized}}$) that is at least as good as the theoretical bound ($P_{\text{bound}}$).
- $P_{\text{centralized}} < P_{\text{bound}}$: The centralized ES model is trained on a smaller dataset from a single source/institution, limiting its ability to capture the full data diversity present in the entire dataset. Thus, $P_{\text{centralized}}$ is likely to be lower than the theoretical bound ($P_{\text{bound}}$).

Empirical validation on datasets and comprehensive experimentation would be essential to draw concrete conclusions about performance comparison between decentralized and centralized models.

### 3.2 AI model predicting early-stage AD

The expert systems are being developed using various techniques, which are mostly used to assist medical practitioners in diagnosis. In this study, we need to train the AI model (Figure 1) via the data that we have obtained from the Alzheimer's Disease Neuroimaging Initiative (ADNI) database (http://adni.loni.usc.edu), a public-private partnership launched in 2003 by Michael Weiner, MD. Our proposed framework consists of processing steps: feature extraction, feature selection, and classification. We examined different feature selection methods to choose an optimal subset of features, maximizing the accuracy of classification between cognitively normal (CN), individuals with significant memory concern (SMC) and mild cognitive impairment (MCI) patients. The subjects are randomly split into training and testing datasets, the classifier is trained using the training dataset, and the testing dataset is passed to the trained classifier to measure the performance.

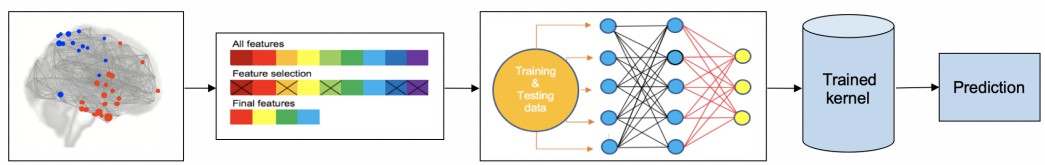

Figure 1: AI classification model

We have used data for 561 subjects total, among those, 231 SMC, 259 CN, and 71 MCI patients. The feature selection algorithms were applied to the graph features (degree centrality for each ROI) to select the most discriminating features for the classification of MCI, SMC, and CN subjects. The Sequential Forward Selection feature selection algorithm and the Random Forest classifier resulted in a satisfying performance with accuracy of more than 92% as shown in Figure 2. We run the models

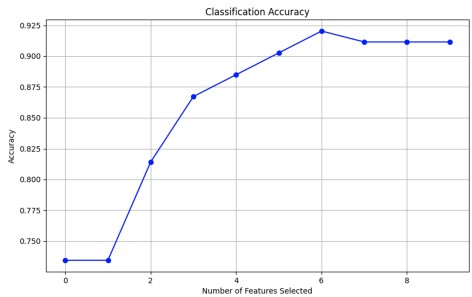

Figure 2: Classification accuracy of AI model.

on a MacBook Pro equipped with an Intel Core i9 processor, featuring 8 cores, speed of up to 4.8 GHz, and 30 GB of RAM.

The graph features were obtained by applying graph theory analysis on rs-fMRI images. The pre-processing, network modeling for graph feature extraction is done via FSL library. The patients can therefore input their MRI images via the provided App, and the FSL library processes, and generates the brain connectivity matrix. From longitudinal measures, patients are labeled as non-convertors and convertors fulfilling the criteria for Prodromal AD's continuum according to Jack et al. (2018). At this stage, we have just trained the AI model with publicly available ADNI data.

### 3.3 Hierarchical Federated Learning

Our initial choice of using federated learning combined with blockchain technology was motivated by the need for decentralized, secure data sharing, and crowdsourcing in healthcare settings (Behfar and Crowcroft, 2024). MRI scans are highly sensitive and specific to individual patients. Pre-trained models, while beneficial for general tasks, may not be optimally suited for such intricate and specific patterns. Using pre-trained models could risk overfitting, potentially compromising patient privacy and the model's generalizability to new, unseen data. Furthermore, diagnosis often requires specific feature representations that capture subtle variations in brain images indicative of the disease. Transfer learning, while effective, might not allow for the fine-tuning required to extract these specialized features optimally.

Implementing a hierarchical federated learning system within a blockchain-based platform for Alzheimer's Disease (AD) diagnosis represents an innovative approach to medical data analysis and privacy preservation. In this setup, patients upload their medical test data, including MRI images onto the generated DApp. This application acts as a gateway to the decentralized platform, leveraging blockchain for data integrity and security (appendix A and B). The hierarchical federated learning process then unfolds in a structured manner across a cluster of nodes, ensuring that patient data remains localized and secure throughout the learning process.

The procedure begins with the division of the federated network into clusters, each corresponding to a specific a group of nodes within the healthcare ecosystem, such as hospitals or research institutions. Within each cluster, local learning models are trained on the patient data available to that cluster. This local training process allows each node to develop an understanding of AD features and indicators based on the subset of data it has access to, without exposing patient data beyond its original location. After local model training, each cluster aggregates its findings to update a local model. The hierarchical aspect of this approach comes into play with the aggregation of these locally updated models across the network. Instead of directly combining data from all nodes, the models trained locally within each cluster are first aggregated to form a cluster-level model. These cluster-level models then contribute to the training of a global model.

### 3.4 Anomaly Detection

There are issues related to bias, data quality and inconsistency in the data collection/labelling, and performing an exhaustive check of the correctness and quality of MRI images and biological information directly on-chain is not practical due to the computational complexity and cost constraints

of blockchain platforms. Instead, such checks are typically performed off-chain, and the blockchain is used to record the results securely.

A practical example of a smart contract that allows patients to submit their data along with a brief initial evaluation is given in Listing 1 (see Appendix C). The contract stores this data on-chain and allows patients to verify and timestamp their submissions. Note that this contract primarily serves as a ledger for the data and initial evaluation results, and more comprehensive checks should be performed off-chain by the application (DApp) before submitting data to the blockchain. In this contract, the "submitCertificate" function allows patients to submit the results of the off-chain anomaly detection process. The "verifyCertificate" function allows patients to verify their certificates. One can implement additional verification steps in the "verifyCertificate" function as needed. To implement a smart certificate for anomaly detection on the client side of a medical data sharing platform, we would use off-chain data analysis techniques since performing anomaly detection directly on-chain can be expensive and inefficient due to the trade-off between performance and security.

**Data Collection:** Patients provide their biological information and MRI images along with timestamps to the application.

**Off-Chain Anomaly Detection:** Implement advanced anomaly detection algorithms off-chain within the App. For MRI images, one might use computer vision techniques, and for biological information, statistical or machine learning methods can be applied to detect anomalies. These algorithms should thoroughly evaluate the correctness and quality of the data.

**Smart Certificate Creation:** After off-chain anomaly detection, create a detailed smart certificate within the App to include:

- Anomaly type (e.g., incorrect data, bad images, etc.).
- Timestamp.
- Metadata about the data and the anomaly.
- Any relevant context or notes about the anomaly.

**Blockchain Interaction:** Use a smart contract on the blockchain to securely store and verify the smart certificates generated within the App. The smart contract records the results of the anomaly detection process, providing an immutable and auditable record.

### 3.4.1 Off-chain anomaly detection for biological information

For biological information, anomaly detection can involve statistical methods or machine learning techniques, depending on the nature and structure of the data. Here in Listing 2 (see Appendix C), we provide an approach using Python and the popular scikit-learn library: In this example, we perform the following steps:

- Load biological data.
- Select the relevant features for anomaly detection.
- Apply feature scaling using StandardScaler.
- Reduce dimensionality using PCA.
- Choose an anomaly detection model (Isolation Forest, or) and fit it to the reduced data.
- Predict anomaly scores for each data point.
- Define a threshold for anomaly detection.
- Identify anomalies based on the threshold.

### 3.4.2 Off-chain anomaly detection for MRI images

Detecting anomalies in MRI images typically involves computer vision techniques and deep learning models. One might consider using popular deep learning libraries like TensorFlow or PyTorch. Here in Listing 3 (see Appendix C), we provide an approach using a pre-trained model. This approach allows to detect anomalies in MRI images based on how well the autoencoder can reproduce the input image. Anomalies will typically result in higher MSE values compared to normal images. One might need to fine-tune the threshold based on the dataset and requirements. In this code:

- Load a pre-trained autoencoder model (both encoder and decoder parts). Autoencoders learn to encode data efficiently and are often used for anomaly detection because they can reproduce normal data accurately.

- Load an MRI image (replace 'mri_image.png') and preprocess it.
- Encode the image using the autoencoder's encoder part, then decode it to get a reconstructed image.
- Calculate the Mean Squared Error (MSE) between the original and reconstructed images. This measures how well the model can reproduce the input.
- Set a threshold for the MSE, above which an anomaly is detected.

## 4 System Development

### 4.1 System Architecture

In regard to System Development status, all the system components according to the class diagram in Figure 3 have already been developed. The user-interface application is based on FSL library, and performs MRI data processing, and will be discussed further in the application development section. The underlying blockchain technology for decentralized data sharing has already been developed, which is based on hyperldger fabaric technology for on-chain, and IPFS for off-chain data sharing as pilot project. There are alternative solutions such as zero-knowledge and optimistic rollups (Behfar et al., 2023). The ML models for early AD detection have also been developed, trained, and tested using public dataset ADNI, mentioned above in "AI Model Predicting Prodromal AD", as shown in algorithm 1. The model is supposed to update or learn from new data in the federated learning setup. Figure 3 illustrates the class diagram of the whole system, where each class is defined below:

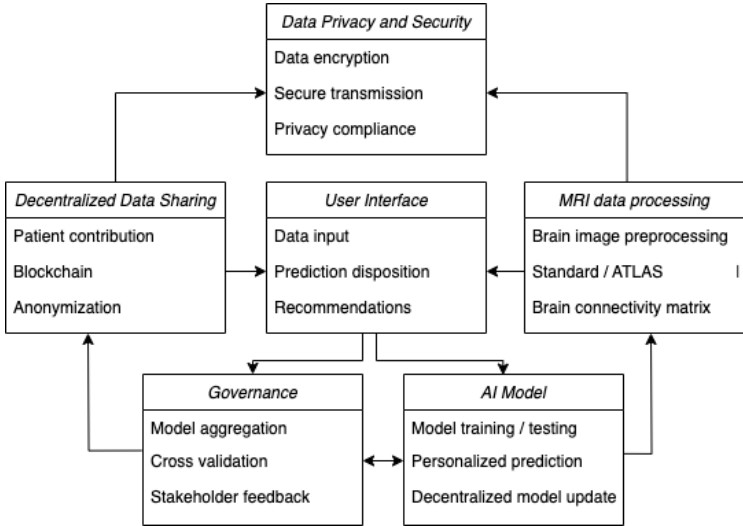

Figure 3: Class diagram

**User Interface:** This is the primary interface for patients to input their anonymous biological info and MRI images and receive prediction deposition and recommendations; this includes approaching a specialist for further and more certain diagnostics. It's the front-end through which users interact with the system.

**Data Security and Privacy:** This component would be responsible for ensuring that patient data, particularly sensitive MRI images, are handled securely and in compliance with privacy regulations. It interfaces with both the User Interface (to ensure that data is securely transmitted) and the Decentralized Data Sharing component (to ensure that data is securely stored and shared).

**MRI Data Processing:** This component processes the MRI images provided by patients through the User Interface. It uses tools like the FSL library for generating brain connectivity matrices, which are crucial for AD prediction. This processed data would then be fed into the AI Model for analysis and prediction/classification.

**Decentralized Data Sharing:** This component is responsible for the secure and anonymous management of patient data within the decentralized network. It ensures that data from various patients is collected without compromising individual privacy.

**AI Model:** The AI model, possibly a Random Forest classifier or similar, is trained on the aggregated

| Algorithm 1: Decentralized expert system for early-stage AD detection. |
|---|

1. **Data Collection and Brain Connectivity Matrix Generation:**
- Patients use the DApp to input their MRI images.
- The FSL library processes the MRI images and generates the brain connectivity matrix.

3. **AI Model Training and Personalization:**
- The AI model, trained initially on a public dataset, can be further fine-tuned and personalized using the brain connectivity matrices.
- The model continuously learns from new patient data, improving its accuracy and adaptability.

2. **Hierarchical Federated Learning:**
- The generated brain connectivity matrices are shared within a local cluster.
- Patients' data may be stored in a privacy-preserving manner, ensuring that the network adheres to privacy regulations.
- The models are updated locally, and parameters are shared globally, aggregated and averaged, and sent back to local clusters.

4. **Prediction and Longitudinal Monitoring:**
- Patients' longitudinal data is used to monitor disease progression over time.
- The trained AI model predicts the transformation to AD based on the inputted MRI images and patient's longitudinal data.

5. **Feedback Loop:**
- Patient feedback and outcomes is collected to improve the model's performance and refine the prediction process. Regular updates based on the latest data and patient feedback ensure that the AI model stays up-to-date and personalized.

brain connectivity matrices. It's responsible for early-stage AD detection, and making predictions about the progression to AD. This model will continuously learn from new patient data, in federated learning or decentralized model update setup, improving its accuracy and adaptability over time.

**Governance:** This component oversees the overall functioning of the system, ensuring that all parts work together cohesively, aggregating model, adhere to set standards and regulations. It will also be involved in updating the system, incorporating patient feedback, and ensuring the system's continuous improvement.

To implement the described decentralized expert system, one needs to integrate several components and consider the role of patients in the system. The overview of the implementation steps is given in Algorithm 1. Regarding the role of patients in the system:

- Patients primarily interact with the system as users. They provide input data, receive predictions, and have access to monitoring and recommendations. They are not typically considered global nodes in the entire decentralized network, but nodes in local clusters.

- The decentralized network consists of nodes that share and process data. These nodes may include user systems, AI model components, and cluster of users.

Our presumed experimentation encompasses several critical scenarios. Firstly, we evaluate the efficiency of the User Interface in terms of data input speed, user satisfaction, and the clarity of prediction results. Secondly, the Data Security and Privacy component's effectiveness will be assessed to ensure the confidentiality and integrity of patient data, checking for potential unauthorized access. The accuracy and reliability of the MRI Data Processing component is tested against benchmarks, assessing the quality of the generated brain connectivity matrices crucial for AD prediction. The system's capability to securely manage patient data within the decentralized network is also be measured, focusing on the speed, efficiency, and security of data sharing and retrieval processes. Moreover, the AI model's performance in early-stage AD detection is validated using metrics such as accuracy, precision, recall, F1-score, and the ROC curve.

## 4.2 Application Development

For our developed application which does MRI preprocessing, we use FSL library which is extremely powerful when it comes to applying and automating workflow since it can unify some of the most crucial steps into one pipeline only. The scripts from the FSL library can be run on either Linux or macOS. FSL unifies some of the most crucial steps into one pipeline only and thereby facilitate the entire workflow, see https://github.com/*****, also note that to use FSLNets either Octave or MATLAB must be running. Putting all the steps together, here is what a workflow could look like:

- Skull stripping – using BET
- Preprocessing – using the modules indicated at the preprocessing step
- Node definition – using MELODIC and Octave
- Generating connectivity matrix – using FSLNets

The backend of this application will not only mange the project's APIs, from frontend to backend to database and vice-versa, but also manage the interaction with FSL and Octave. The latter is indispensable for the creation of the Brain Connectivity Matrix (BCM). As indicated in Figure 4, Schlappinger (2023), all user requests always pass via the server's API-service first, and are dispatched to the corresponding service. When the user tries to log in, the log-in data is sent to the backends' API service, then sends it to the corresponding application service, which in this case would be the authentication service. It handles the transferred data and asks for identification by sending requests to the database. The database response is sent to the application service, and the response back to the API. With the definition of the expert system, the web application does preprocessing on the subjects to finally output the brain connectivity matrix that is available immediately after processing.

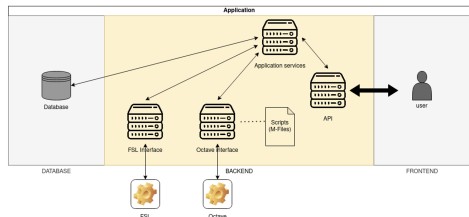

Figure 4: backend-frontend infrastructure diagram.

In terms of scalability, our system is designed to efficiently manage and process large volumes of patient data, making it highly scalable to accommodate the growing demands of medical data analytics. The decentralized architecture leverages blockchain technology, specifically Hyperledger Fabric for on-chain data storage and IPFS for off-chain data sharing, ensuring secure and distributed data management across the network. This decentralized approach allows the system to seamlessly integrate new patient data sources without imposing significant overhead or compromising data privacy. Moreover, the federated learning setup enables collaborative model training across multiple nodes, allowing the AI model to learn from diverse and geographically distributed datasets while preserving data locality and reducing computational burden on individual nodes. Additionally, the modular design of the system as depicted in the class diagram (Figure 3) facilitates independent scaling of each component enabling efficient resource allocation and optimal performance even as the system expands to incorporate more patients, data sources, and computational nodes. Thus, our system not only ensures data security and privacy but also exhibits high scalability and efficiency.

# 5 Conclusion

In this paper, we have presented a novel approach to address the challenges associated with managing and analyzing massive centralized repositories of MRI data and persistent privacy concerns for early AD prediction. Our primary position advocates for the integration of blockchain technology with federated learning to establish a decentralized expert system. This system aims to preserve data privacy, ensure security, and facilitate efficient analysis across decentralized network. Overall, the decentralized expert system for early-stage AD detection can leverage the decentralized collected data and intelligence to provide accurate and timely predictions. Our expert system serves as a model tool that collects patients' data in a decentralized way via our FSL-built application. FSL using Octave creates brain connectivity matrices and pushes to the AI engine. Our trained model uses Sequential Forward Selection feature selection algorithm and the Random Forest classifier resulting in accuracy of more than 92%; the classification model is retrained by updated parameters based on hierarchical federated learning setup. This method offers a scalable, privacy-preserving framework for leveraging vast amounts of medical data, potentially leading to more accurate and early detection of AD, while ensuring patient data remains secure and private. This not only helps individuals to detect early-stage AD in time, but also helps clinics and hospitals who are willing to use this solution to effectively monitor the patients and predict their progression with less ambiguity.

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

# A    Application security

Ensuring the security and privacy of medical data is of paramount importance in our system development. We implement a comprehensive set of measures to safeguard sensitive information, maintain data integrity, and comply with privacy regulations.

**Data Encryption**
End-to-End Encryption: All medical data, including biological information and MRI images, undergo end-to-end encryption using industry-standard encryption algorithms. This means that data is encrypted at its source (on the patient's side) and remains encrypted during transmission and storage within our system. Even if an unauthorized entity intercepts the data, it remains indecipherable without the encryption keys.

AES Encryption: We employ the Advanced Encryption Standard (AES) for data encryption. AES is a widely recognized and robust encryption algorithm known for its security and performance. It ensures that patient data is protected from unauthorized access.

**Secure Transmission**
HTTPS: We utilize the Hypertext Transfer Protocol Secure (HTTPS) for web-based data transmission. HTTPS is a secure communication protocol that combines the standard HTTP with encryption using Transport Layer Security (TLS) or Secure Sockets Layer (SSL) protocols. This encryption layer ensures that data exchanged between the client and our system is shielded from eavesdropping and tampering during transit.

Blockchain Technology: Our system leverages blockchain technology to enhance the security of data sharing. Blockchain, with its decentralized and immutable ledger, provides an additional layer of protection. Each data transaction is recorded on the blockchain, and once added, it cannot be altered. This ensures transparent and secure data sharing among authorized parties.

**Privacy Compliance**
Access Control: Access control mechanisms are in place to restrict data access to only authorized healthcare professionals and patients. Role-based access control ensures that individuals can only access the data that is relevant to their responsibilities. Patients have control over who can access their data, granting consent for sharing, and revoking access as needed.

HIPAA and GDPR Compliance: Our system adheres to the Health Insurance Portability and Accountability Act (HIPAA) and the General Data Protection Regulation (GDPR), in addition to local data protection laws. These compliance measures provide a legal framework for the secure handling of patient data, including rules for data access, storage, and sharing.

Regular Audits and Privacy Impact Assessments: To maintain compliance, we need to conduct regular system audits and privacy impact assessments. These evaluations help us identify and rectify potential privacy issues and vulnerabilities in our system. They also ensure that we remain aligned with the latest data protection regulations.

Even if patient data is anonymized, it's often advisable and may be legally required to comply with many of the security and privacy measures mentioned above. Anonymization can reduce the risk associated with the disclosure of sensitive information, but it doesn't necessarily exempt a system from all privacy regulations or security best practices.

# B  Scope Limitations and Societal Impact

Despite the promising aspects of the proposed system, several limitations need to be acknowledged:

- Data Quality and Consistency: The accuracy of the AI model heavily relies on the quality and consistency of the input data. Variability in MRI image quality, biological information, and other patient-contributed data can affect the model's performance.

- Computational Complexity: Performing exhaustive checks of MRI images and biological data directly on the blockchain is not feasible due to the high computational costs. This necessitates off-chain processing, which may introduce additional complexity and potential delays.

- Model Generalizability: The AI model is initially trained on public datasets, which may not fully capture the diversity of the broader patient population. While the system can update the model with new patient data, initial predictions might be less accurate for underrepresented groups.

- Privacy and Security Concerns: Although blockchain technology enhances data security and privacy, it also introduces new challenges. Ensuring that all aspects of patient data handling comply with privacy regulations and maintaining robust security measures against potential cyber threats are ongoing concerns.

- Technical Barriers for Patients: The decentralized nature of the system requires patients to engage with technology such as blockchain wallets and data submission interfaces. This could be a barrier for less tech-savvy individuals, potentially limiting the system's accessibility and usability.

- Regulatory and Ethical Issues: The deployment of such a decentralized medical diagnostic system must navigate complex regulatory landscapes. Ensuring compliance with medical standards, obtaining necessary approvals, and addressing ethical considerations related to AI-driven medical predictions are critical challenges.

- Scalability: As the number of users and the volume of data increase, the system's scalability could become a concern. Efficiently managing large datasets and ensuring timely processing and predictions in a decentralized environment will require ongoing optimization.

The development and implementation of a decentralized expert system for early-stage Alzheimer's disease prediction hold significant societal implications. On the positive side, this technology promises to enhance early detection and intervention, leading to improved patient outcomes and quality of life. By enabling timely and accurate predictions, patients can benefit from early treatment, potentially slowing disease progression and delaying severe symptoms. The system's use of blockchain technology ensures robust data privacy and security, fostering patient trust in the confidentiality of their health information. Additionally, the ability to update and personalize the AI model with new patient data allows for more tailored healthcare solutions, offering personalized treatment plans that cater to individual needs. This, in turn, can reduce long-term healthcare costs by decreasing the need for intensive care in advanced stages of Alzheimer's disease. Moreover, the secure sharing of anonymized data for research purposes can accelerate scientific discoveries and the development of new treatments.

However, the deployment of such a system also presents challenges. The reliance on digital tools for data submission and interaction may exclude individuals who lack access to technology or have limited digital literacy, potentially exacerbating health disparities among older adults and socioeconomically disadvantaged groups. Despite blockchain's security measures, there may still be privacy concerns, and any data breaches could undermine patient trust. Ethical and regulatory challenges arise from the need to ensure the accuracy and fairness of AI-driven predictions, and obtaining necessary approvals remains an ongoing hurdle. Over-reliance on technology might marginalize human clinical expertise, highlighting the importance of maintaining a balance between AI support and healthcare professional judgment. Additionally, the economic implications of implementing and maintaining such advanced systems must be considered, as they may impose financial burdens on healthcare providers and patients. By addressing these societal impacts thoughtfully, the deployment of the decentralized expert system can maximize its benefits while minimizing potential harms, contributing to more equitable and effective healthcare.

## C  Listings

Here are all the referred listings in the main text:

The smart contract, listing 1 named MedicalDataSubmission, enables patients to securely submit and verify their medical data on the Ethereum blockchain. The contract defines a PatientData structure that includes the patient's address, biological information, evaluation, timestamp, and a verification status. Patients can submit their data using the submitData function, which ensures that both the biological information and evaluation are non-empty before storing the data along with the current timestamp and an initial unverified status. The submitted data is added to the submissions array, and an event DataSubmitted is emitted to log the submission details. Patients can later verify their own submissions using the verifySubmission function, which checks that the submission exists, the caller is the patient who submitted the data, and the submission has not already been verified. Upon successful verification, the submission's status is updated to verified. This contract ensures data integrity and provides a transparent mechanism for patients to manage their medical information.

---

Listing 1: A smart contract that allows patients to submit data and verify.

```
// SPDX-License-Identifier: MIT
pragma solidity ^0.8.0;
contract MedicalDataSubmission {
    struct PatientData {
        address patient;
        string biologicalInfo;
        string evaluation;
        uint256 timestamp;
        bool isVerified;
    }
    PatientData[] public submissions;
    event DataSubmitted(uint256 indexed submissionId,
    address indexed patient, string biologicalInfo,
    string evaluation, uint256 timestamp);

    function submitData(string memory biologicalInfo,
    string memory evaluation) external {
        require(bytes(biologicalInfo).length > 0,
        "Biological information cannot be empty.");
        require(bytes(evaluation).length > 0,
        "Evaluation cannot be empty.");
        submissions.push(PatientData(msg.sender,
        biologicalInfo, evaluation, block.timestamp, false));
        uint256 submissionId = submissions.length - 1;
        emit DataSubmitted(submissionId, msg.sender,
        biologicalInfo, evaluation, block.timestamp);
    }

    function verifySubmission(uint256 submissionId) external {
        require(submissionId < submissions.length,
        "Submission does not exist.");
        PatientData storage submission =
        submissions[submissionId];
        require(msg.sender == submission.patient,
        "Only the patient can verify the submission.");
        require(!submission.isVerified,
        "Submission is already verified.");
        // Implement additional verification steps as needed
        submission.isVerified = true;
    }
}
```

---

The code, listing 2, demonstrates the process of anomaly detection in biological data using the Isolation Forest algorithm. It begins by loading the biological data and selecting relevant features

for anomaly detection. The selected features are scaled using StandardScaler to normalize the data. To reduce dimensionality and highlight the most significant features, Principal Component Analysis (PCA) is applied, transforming the data into a two-dimensional space. The Isolation Forest model, designed to detect anomalies, is trained on this transformed data, with a contamination rate of 5% indicating the expected proportion of anomalies. Anomaly scores are calculated for each data point, and a threshold is set to identify anomalies. Data points with scores below this threshold are flagged as anomalies. The script prints the details of the detected anomalies for further analysis. Additionally, it encourages experimenting with other anomaly detection models like Elliptic Envelope and One-Class SVM, and fine-tuning parameters to enhance detection performance.

---

Listing 2: Anomaly detection for biological information using Isolation Forest.

```python
import numpy as np
from sklearn.preprocessing import StandardScaler
from sklearn.decomposition import PCA
from sklearn.covariance import EllipticEnvelope
from sklearn.ensemble import IsolationForest
from sklearn.svm import OneClassSVM

# Load your biological data
biological_data = load_biological_data()

# Select the relevant features for anomaly
detection
selected_features = ['feature1', 'feature2',
'feature3']
X = biological_data[selected_features]

# Apply feature scaling
scaler = StandardScaler()
X_scaled = scaler.fit_transform(X)

# Apply dimensionality reduction using PCA
pca = PCA(n_components=2)
X_pca = pca.fit_transform(X_scaled)

# Choose an anomaly detection model
model = IsolationForest(contamination=0.05)
model.fit(X_pca)

# Predict anomalies
anomaly_scores = model.decision_function(X_pca)

# Define a threshold for anomaly detection
threshold = -0.3  # Adjust as needed

# Identify anomalies
anomalies = biological_data[anomaly_scores
< threshold]

# Further processing or reporting of anomalies
for index, row in anomalies.iterrows():
    print(f"Anomaly detected for sample {index}
    :")
    print(row)

# experiment with different models
# (Elliptic Envelope, One-Class SVM, etc.)
# and fine-tune parameters for better anomaly
detection performance.
```

---

The code snippet, listing 3, demonstrates an off-chain method for anomaly detection in MRI images using a pre-trained autoencoder model. The process begins by loading the pre-trained autoencoder model, followed by loading and normalizing an MRI image. The image is then preprocessed to match the input size required by the model, which includes resizing the image and adding a batch dimension. The autoencoder encodes the image and subsequently reconstructs it. The Mean Squared Error (MSE) between the original and reconstructed images is calculated as the reconstruction loss. An anomaly is detected if this loss exceeds a predefined threshold (set to 0.01 in this example), indicating that the MRI image significantly deviates from the normal patterns learned by the autoencoder. Depending on the reconstruction loss, the script outputs whether an anomaly is detected or not.

Listing 3: Off-chain anomaly detection for MRI images.

```python
import tensorflow as tf
import numpy as np
from PIL import Image

# Load pre-trained autoencoder model
autoencoder = tf.keras.models.load_model
('autoencoder_model.h5')

# Load an MRI image
image = Image.open('mri_image.png')
# Normalize image data
image = np.array(image) / 255.0

# Preprocess the image for model input
# Resize to the model's input size
input_image = tf.image.resize(image, (224, 224))
# Add batch dimension
input_image = np.expand_dims(input
_image, axis=0)

# Encode the image using the autoencoder
encoded_image = autoencoder.encoder(input_image)
.numpy()

# Calculate reconstruction loss
reconstructed_image = autoencoder(input
_image).numpy()
mse = np.mean(np.square(input_image -
reconstructed_image))

# Define a threshold for anomaly detection
threshold = 0.01  # Adjust as needed

if mse > threshold:
    print("Anomaly detected in MRI image.")
else:
    print("No anomaly detected in MRI image.")
```

