# OpenReview forum: "Architecture of Decentralized Expert System for Early Alzheimer's Prediction Enhanced by Data Anomaly Detection"
_NeurIPS.cc/2024/Conference — Submitted to NeurIPS 2024_

### Official Review · Reviewer_ahuS · 2024-07-08

**Soundness:** 2
**Presentation:** 1
**Contribution:** 2
**Rating:** 3
**Confidence:** 4

**Summary:**

This work introduces a novel approach to diagnosing Alzheimer's Disease using a decentralized expert system. This system leverages blockchain technology and Federated Learning to enhance data privacy and manage large volumes of MRI data effectively. The key innovation lies in integrating these technologies to address the challenges of traditional diagnostic methods, which often suffer from delays and inaccuracies, especially in the early stages of the disease.

**Strengths:**

1. This work presents a pioneering integration of blockchain technology and Federated Learning to enhance Alzheimer's Disease (AD) diagnostics, addressing privacy concerns and data management challenges.
2. The proposed decentralized expert system architecture, which includes anomaly detection for patient-submitted data, showcases a comprehensive approach to AD diagnostics, emphasizing AI-driven MRI analysis.

**Weaknesses:**

1. While the system shows promising results, the article does not provide extensive comparative data against traditional centralized systems or other decentralized approaches, which could validate its superiority more robustly. This work lacks of comparative performance data.
2. The complexity of the blockchain and Federated Learning components might pose usability challenges for less technically adept users, potentially affecting the system's adoption.
3. There are no more details of the algorithms this work used, maybe give out more meaningful algorithm design for the specific model you are using.

**Questions:**

How well does the AI model generalize to diverse populations, given the variability in MRI data quality and the limited scope of initial training datasets?

**Limitations:**

1. The accuracy of the AI model heavily depends on the quality and consistency of input data, which might vary significantly across different healthcare settings.
2. The use of advanced technologies such as blockchain might limit the accessibility of the system for users not familiar with such technology, potentially restricting its applicability.

---

> ### Author Rebuttal · Authors · 2024-08-04
>
> - While the system shows promising results, the article does not provide extensive comparative data against traditional centralized systems or other decentralized approaches, which could validate its superiority more robustly. This work lacks of comparative performance data.
>   - Response: The current version of the paper primarily introduces the architectural framework and theoretical benefits of our proposed system. The focus has been on detailing the integration of blockchain technology and federated learning to enhance privacy and scalability in medical data processing. Future publications will explore: User Interaction, Real-world Implementation Challenges, Use-Case Elaboration. Although this is a pure architecture paper, we have trained an AI model where we used data for 561 subjects total, among those, 231 SMC, 259 CN, and 71 MCI patients. The feature selection algorithms were applied to the graph features (degree centrality for each ROI) to select the most discriminating features for the classification of MCI, SMC, and CN subjects. Initial AI model is generated to be deployed on the nodes; and we plan to undertake a comprehensive experiment designed to validate Federated Learning with new patients' data acquired via DApp. This will involve testing the model across different demographics and disease stages to statistically ascertain the impact of data diversity on prediction accuracy. See the list of the major points ABOVE.
> - The complexity of the blockchain and Federated Learning components might pose usability challenges for less technically adept users, potentially affecting the system's adoption.
>   - Thank you for raising this important concern regarding the usability challenges that could arise from the integration of complex technologies such as blockchain and Federated Learning in our system. Ensuring ease of use is critical for the adoption and effectiveness of any new technology, especially in a healthcare context where users range from highly technical staff to patients who may not have extensive technological expertise.
> For a practical example of a blockchain system where patients are reimbursed with Ether for sharing anonymous medical tests, we refer to the study (https://doi.org/10.1017/dap.2024.4). which provides insights into the incentives and data handling aspects that are relevant to our proposed architecture.
> Wherever possible, complex processes, especially those related to data handling and model training, will be automated.
> Incorporating feedback mechanisms within the system will allow us to continuously gather user experiences and identify aspects of the system that are particularly challenging for users.
> - There are no more details of the algorithms this work used, maybe give out more meaningful algorithm design for the specific model you are using.
> - Response: Here's an outline of the specific algorithms used and how they contribute to the functionality of our model:
>   - Blockchain Algorithm: The system employs Ethereum smart contracts for handling data submissions, access control, and transactions. Smart contracts automate these processes, ensuring they are executed under predefined conditions without third-party intervention. See Listings 1,2,3 - smart contracts used for data anomaly detection.
>   - Federated Learning Algorithm: - Model Initialization: The global model is initialized using a lightweight SVM, suitable for processing medical imaging data. The initialization parameters are optimized to balance training speed and model accuracy. - Local Training: At each node, local model updates are performed using Stochastic Gradient Descent (SGD) with backpropagation. To accommodate diverse data distributions at different nodes, we implement adaptive learning rates.- Aggregation Protocol: We employ Federated Averaging (FedAvg) for aggregating local model updates. See Algorithm 1.
>   - Anomaly Detection Algorithm: For pre-processing data, we utilize the Isolation Forest algorithm to identify and isolate anomalies in the dataset (The code, listing 2, demonstrates the process of anomaly detection in biological data using the
> Isolation Forest algorithm). This step is crucial for maintaining the quality of data used in training the Federated Learning model.
> - How well does the AI model generalize to diverse populations, given the variability in MRI data quality and the limited scope of initial training datasets?
> - Response: Thank you for raising a critical question regarding the generalizability of our AI model to diverse populations. This is particularly pertinent given the inherent variability in MRI data quality and the constraints of the initially available training datasets.
>   - Model Initialization and Training: - Lightweight Initialization: The AI model is initiated using a lightweight architecture designed to be responsive and efficient. The initialization parameters of the model are carefully optimized to balance training speed with accuracy. - Local Training Adaptation: During local training, each participating node adjusts the model using its local data, which likely includes diverse patient demographics and varying MRI data quality. This local adaptation helps tailor the model to specific sub-populations.
>   - Handling Data Quality Variability: - Off-Chain Data Quality Verification: To address concerns regarding MRI data quality, we employ an off-chain verification process through smart contracts on each edge device (Listings 2,3). This process ensures that only high-quality, verified data is used during the training phase. By filtering out poor quality or anomalous data early in the process, we maintain the integrity and reliability of the training data. - Robustness to Data Variability: The federated learning framework inherently enhances the model's robustness to data variability. By aggregating diverse local updates, the model learns to generalize across a broader spectrum of data characteristics than if trained on a homogenized central dataset.

---

> > ### Comment · Reviewer_ahuS · 2024-08-11
> > **Reply to Rebuttal by Authors**
> >
> > Thank you for your response, it is still not clear that if this proposed framework for Alzheimer's is useful which somehow should be proved by comprehensive experiments, and my concerns about the missing details of algorithms of each part (such as blockchain, federated learning, anomaly detection) haven't been addressed based on your response, which is a kind of definition but not formulas for explaining your algorithms. The most important thing you need to do is to prove your framework by experiments (at least doing experiments with simple algorithms, small datasets).

---

> ### Author Response · Authors · 2024-08-11
>
> Thank you for your insightful feedback and the emphasis on the importance of experimental validation. I appreciate the opportunity to clarify the intent and scope of our current manuscript, which primarily aims to introduce an innovative architectural framework for a decentralized, privacy-preserving system tailored for Alzheimer’s disease prediction
> - see the innovations ABOVE:
>   - Decentralized Data Crowdsourcing with Compensation
>   - Hierarchical Clustering in Federated Learning
>   - Integration of Blockchain for Data Integrity and Auditability
>   - Focus on Early Detection of Alzheimer’s Disease
>   - Unique Application to Alzheimer’s Disease Prediction and Monitoring
>
> - Clarification of Scope:
>   - Architectural Focus: This paper is designed as a conceptual and architectural introduction to a novel framework integrating blockchain technology, federated learning, and advanced data analytics for medical data processing. The primary contributions are the architectural innovations and the theoretical underpinning of how these components interact within our proposed system.
>   - Stage of Research: As an architectural paper, the current focus is on detailing the system design and the potential impacts of its implementation. It establishes the groundwork for future empirical research, which will rigorously test and validate the system’s performance across various metrics. In another study, we have shown how patients being reimbursed for medical test data sharing https://doi.org/10.1017/dap.2024.4. We have also designed a user-friendly app (https://github.com/stefankam/predprodalzheimer), which receives MRI images, change to connectivity matrix and do prediction of stage of Alzheimer without transferring data to any central location.
>
> - Addressing Algorithmic Details:
>   - While comprehensive algorithmic formulas are typically more relevant in an implementation or experimental paper, we include detail about the algorithms’ roles and interactions within the system to provide clarity on how each component contributes to the overall functionality, see the class diagram. Although this is a pure architecture paper, we have trained an AI model where we used data for 561 subjects total, among those, 231 SMC, 259 CN, and 71 MCI patients. The feature selection algorithms were applied to the graph features (degree centrality for each ROI) to select the most discriminating features for the classification of MCI, SMC, and CN subjects. Initial AI model is generated to be deployed on the blockchain nodes. See https://arxiv.org/pdf/1808.03949
>
> Your Question: The complexity of the blockchain and Federated Learning components might pose usability challenges for less technically adept users, potentially affecting the system's adoption.
> - Response: While it is true that advanced technologies such as blockchain might pose a learning curve for some users, the primary intention of our system is not to require widespread adoption of blockchain technology by all patients. Instead, the system is designed with several key features that address accessibility and user engagement:
>   - User Experience Design: The core functionality of our system is designed to be user-friendly and intuitive (https://github.com/stefankam/predprodalzheimer). Patients interact with the application primarily through a straightforward interface that does not require them to have a deep understanding of blockchain technology. The blockchain operates in the background to ensure data integrity and security, while patients focus on interacting with the system through familiar processes, such as submitting MRI data and receiving predictions.
>   - Incentive Mechanism: To encourage participation and ensure engagement, the system includes an incentive mechanism where patients receive Ether in exchange for sharing their anonymous test data. This approach provides a tangible benefit to users, which can help overcome any potential hesitation or lack of familiarity with the underlying technology. By offering financial incentives, we aim to motivate users to participate actively without needing to understand the technical aspects of blockchain.
>   - Blockchain as a Service: Moreover, the blockchain component of our system is implemented as a service that abstracts the complexity away from the end users. This means that users interact with a simplified front-end application, while the blockchain infrastructure manages the data submission, validation, and security behind the scenes. The integration of blockchain technology is intended to enhance data privacy and security, without placing undue burden on the users.
>   - Targeted Application: It is important to note that the system is targeted towards a specific use case where patients are aware of and are motivated by the benefits of participating in a decentralized data-sharing network. The goal is not to achieve universal adoption but to provide a secure and incentivized platform for those who choose to contribute their data.

---

### Official Review · Reviewer_vAAp · 2024-07-11

**Soundness:** 1
**Presentation:** 1
**Contribution:** 1
**Rating:** 2
**Confidence:** 4

**Summary:**

The authors assume that applying blockchain platforms to combine datasets for Alzheimer’s Disease and then using federated learning for multi-centralized training can improve diagnostic performance. However, the manuscript lacks technical details and experimental evidence. All descriptions are conceptual, making the manuscript a proposal rather than a technical paper.

**Strengths:**

It is interesting to apply blockchain platforms to combine datasets for Alzheimer’s Disease and then using federated learning for multi-centralized training.

**Weaknesses:**

There are no technical details and no experiments. Details can be found in Questions and Limitations.

**Questions:**

1.	What is your main contribution, a model, a framework or just a proposal?
2.	How the dataset collected on blockchain platforms?  please make the description in clear.
3.	How you prove the advantage of proposed architecture?  there are no experiments in the manuscript.
4.	How is the federated model trained and what the performance is?

**Limitations:**

1.	No technical details and experiments.
2.	The literature review of Alzheimer’s Disease diagnosis is not complete, especially regarding AI-based approaches.

---

> ### Author Rebuttal · Authors · 2024-08-04
>
> - What is your main contribution, a model, a framework or just a proposal?
> - Response: Our main contribution is the development of a framework. This framework outlines a decentralized expert system architecture that integrates advanced technologies such as blockchain and federated learning for early-stage Alzheimer's disease prediction. This architecture is designed to enhance the security, privacy, and scalability of handling sensitive medical data, while also utilizing anomaly detection mechanisms to ensure the quality and integrity of the data used for AI-driven predictions. Here’s a breakdown of how this contribution can be framed:
>   - Decentralized Architecture: The framework proposes a novel approach by using a decentralized system that leverages blockchain technology. This is intended to ensure data integrity, transparency, and security in the processing and handling of sensitive medical data. See the list of the major points ABOVE.
>   - Integration of Technologies: The integration of blockchain with federated learning within the framework allows for a collaborative, yet secure, approach to developing AI models. See the list of the major points ABOVE.
>   - Anomaly Detection: The framework includes an anomaly detection mechanism to ensure the quality of the data that feeds into the AI models. This is crucial in medical applications where data quality directly impacts the accuracy and reliability of disease predictions.
>   - Scalability and Adaptability: The proposed framework is designed to be scalable and adaptable across different healthcare settings.  In our system, blockchain technology is primarily utilized for two main purposes: facilitating transactions of tokens in exchange for medical data contributions and maintaining a secure and immutable record of data transactions and verification statuses. This focused use of blockchain allows us to leverage its benefits—namely, enhanced security and transparency—without significantly adding to the computational load typically associated with blockchain operations such as complex consensus mechanisms or the handling of large-scale data computations.
> - How the dataset collected on blockchain platforms? please make the description in clear.
>   - For a practical example of a blockchain system where patients are reimbursed with Ether for sharing anonymous medical tests, we refer to the study (https://doi.org/10.1017/dap.2024.4). Which provides insights into the incentives and data handling aspects that are relevant to our proposed architecture.
> A smart contract that allows patients to submit their data along with a brief initial evaluation is given in Listing 1 (see Appendix C). The contract stores this data on-chain and allows patients to verify and timestamp their submissions. Note that this contract primarily serves as a ledger for the data and initial evaluation results, and more comprehensive checks should be performed off-chain by the application (DApp) before submitting data to the blockchain. In this contract, the "submitCertificate" function allows patients to submit the results of the off-chain anomaly detection process. The "verifyCertificate" function allows patients to verify their certificates. One can implement additional verification steps in the "verifyCertificate" function as needed. To implement a smart certificate for anomaly detection on the client side of a medical data sharing platform, we would use off-chain data analysis techniques since performing anomaly detection directly on-chain can be expensive and inefficient due to the trade-off between performance and security.
> - How you prove the advantage of proposed architecture? there are no experiments in the manuscript.
> - Response: Thank you for highlighting the importance of empirical validation in demonstrating the advantages of our proposed decentralized architecture. Although this is a pure architecture paper, we have trained an AI model where we used data for 561 subjects total, among those, 231 SMC, 259 CN, and 71 MCI patients. The feature selection algorithms were applied to the graph features (degree centrality for each ROI) to select the most discriminating features for the classification of MCI, SMC, and CN subjects. Initial AI model is generated to be deployed on the nodes; and we plan to undertake a comprehensive experiment designed to validate Federated Learning with new patients' data acquired via DApp. This will involve testing the model across different demographics and disease stages to statistically ascertain the impact of data diversity on prediction accuracy.
> - How is the federated model trained and what the performance is?
> - Response: Federated Model Training Process. See Algorithm 1
>   - Initialization: A central server initializes a global model with pre-defined architecture suitable for the specific medical prediction task.
>   - Distribution of Model: The initialized model is distributed to participating nodes.
>   - Local Training: Each node trains the model on its local dataset. This training is typically performed using traditional machine learning algorithms, adapted to the specificities of the data (e.g., types of MRI images, patient demographics).
>   - Aggregation:  - After training, each node sends only the model updates (e.g., weights, gradients) to the central server. The data itself remains at the node, preserving privacy. - The central server aggregates these updates to update the global model. This aggregation can be done using methods like weighted averaging or more sophisticated approaches depending on the algorithm used (e.g., Federated Averaging).
>   - Iteration: The updated global model is sent back to the nodes for further training. This process repeats for several iterations until the model converges or meets certain performance criteria.
>   - Once the model training is complete and it has converged satisfactorily, the final model can be deployed for use in predictions. Performance can be evaluated via Accuracy Metrics.

---

> ### Comment · Reviewer_vAAp · 2024-08-13
>
> Thank the authors for the detailed response. I am still not convinced by the 'experiments and results' provided in the rebuttal. I do expect more complete experiments and comparisons in such kind paper. Also the literature is still thin. I would not be able to raise my score.

---

> > ### Author Response · Authors · 2024-08-13
> >
> > Thank you for your insightful feedback and the emphasis on the importance of experimental validation. I appreciate the opportunity to clarify the intent and scope of our current manuscript, which primarily aims to introduce an innovative architectural framework for a decentralized, privacy-preserving system tailored for Alzheimer’s disease prediction
> > - see the innovations ABOVE:
> >   - Decentralized Data Crowdsourcing with Compensation
> >   - Hierarchical Clustering in Federated Learning
> >   - Integration of Blockchain for Data Integrity and Auditability
> >   - Focus on Early Detection of Alzheimer’s Disease
> >   - Unique Application to Alzheimer’s Disease Prediction and Monitoring
> >
> > - Clarification of Scope:
> >   - Architectural Focus: This paper is designed as a conceptual and architectural introduction to a novel framework integrating blockchain technology, federated learning, and advanced data analytics for medical data processing. The primary contributions are the architectural innovations and the theoretical underpinning of how these components interact within our proposed system.
> >   - Stage of Research: As an architectural paper, the current focus is on detailing the system design and the potential impacts of its implementation. It establishes the groundwork for future empirical research, which will rigorously test and validate the system’s performance across various metrics. In another study, we have shown how patients being reimbursed for medical test data sharing https://doi.org/10.1017/dap.2024.4. We have also designed a user-friendly app (https://github.com/stefankam/predprodalzheimer), which receives MRI images, change to connectivity matrix and do prediction of stage of Alzheimer without transferring data to any central location.
> >
> > - Addressing Algorithmic Details:
> >   - While comprehensive algorithmic formulas are typically more relevant in an implementation or experimental paper, we include detail about the algorithms’ roles and interactions within the system to provide clarity on how each component contributes to the overall functionality, see the class diagram. Although this is a pure architecture paper, we have trained an AI model where we used data for 561 subjects total, among those, 231 SMC, 259 CN, and 71 MCI patients. The feature selection algorithms were applied to the graph features (degree centrality for each ROI) to select the most discriminating features for the classification of MCI, SMC, and CN subjects. Initial AI model is generated to be deployed on the blockchain nodes. See https://arxiv.org/pdf/1808.03949

---

### Official Review · Reviewer_9nGY · 2024-07-12

**Soundness:** 1
**Presentation:** 2
**Contribution:** 2
**Rating:** 3
**Confidence:** 5

**Summary:**

The paper presents a decentralized expert system designed to predict early-stage Alzheimer's Disease using AI-driven MRI analysis. The system leverages blockchain technology and Federated Learning to ensure data privacy and security while performing anomaly detection on patient-submitted data. The architecture includes a Web3 application for patients to upload biological information and MRI images securely. The decentralized approach aims to improve early detection and intervention for Alzheimer's Disease, providing a more comprehensive representation of AD patterns and enhancing model performance through data diversity.

**Strengths:**

The paper encapsulates a few novel ideas. They can be summarized as follows:

1. Handling the security and sensitivity of patient medical information is of paramount importance. The authors were motivated by a very relevant problem and presented an approach to blockchain technology with stated aim of providing robust data privacy and security. By building on decades on research on this topic, this approach has the potential to be extended in future with the general updates in this domain.
2. While there are some confusion around their use case (see weakness below), the authors leveraging Federated Learning and a decentralized system to mitigate the challenges associated with model training on centralized data repositories, such as data bottlenecks and privacy concerns
3. The system aims to provide early-stage prediction of Alzheimer's Disease, which is crucial for timely intervention and improved patient outcomes.

**Weaknesses:**

However given the commendable motivations there are several challenges with the current paper,

1. First and perhaps the most important aspect is that the paper fails to present the real-world challenges associated with the adoption of such decentralized approaches, especially as it pertains to patients engaging with blockchain wallets and data submission interfaces. Also, the primary use-case for the decentralized approach is not evident - is model training the prime use-case or is the main use case patients being able to generate inferences on their own medical records. Overall, the usage scenario around the setup needs to be better motivated and established
2. The paper also lacks formalism around the presentation. For example, if the primary contribution is the architecture around the decentralized AI approach, the design principles needs to be better justified and articulated. A system architecture diagrams needs to be established as well. Similarly, the "proof" around the decentralized approach is not a rigorous mathematical proof. Rather the logic is derived from a hypothesis that more diverse data should lead to a better model. This is a hypothesis at the best and needs to be experimentally validates
3. Finally, the paper is lacking in experimental validation. For example, the proof needs to be backed by real world experiments. Also, this is not the first paper to posit a federated learning approach to medical AI prediction. Some of the SOTA methods in this space needs to be compared against

**Questions:**

Please see the weakness above

**Limitations:**

Please see the weakness above

---

> ### Author Rebuttal · Authors · 2024-08-04
>
> - First and perhaps the most important aspect is that the paper fails to present the real-world challenges associated with the adoption of such decentralized approaches, especially as it pertains to patients engaging with blockchain wallets and data submission interfaces.
> - Response: The current paper introduces a novel architecture for a decentralized AI system tailored for early Alzheimer’s disease prediction, which multiple applied studies will be built. Future publications will explore: User Interaction, Real-world Implementation Challenges, Use-Case Elaboration
>   - In this initial phase, we outline multiple potential use-cases to showcase the versatility of the architecture. The primary use-case will indeed depend on specific implementation strategies, which will be the subject of our next series of papers. This includes a deeper exploration into whether model training or patient-driven inferences provide the most utility in practical scenarios.
>   - For a practical example of a blockchain system where patients are reimbursed with Ether for sharing anonymous medical tests, we refer to the study (https://doi.org/10.1017/dap.2024.4). which provides insights into the incentives and data handling aspects that are relevant to our proposed architecture.
> - The paper also lacks formalism around the presentation. For example, if the primary contribution is the architecture around the decentralized AI approach, the design principles needs to be better justified and articulated. A system architecture diagrams needs to be established as well. Similarly, the "proof" around the decentralized approach is not a rigorous mathematical proof.
> - Response: Our paper primarily introduces a novel architecture for a decentralized AI system tailored for early Alzheimer’s disease prediction. The presentation style was chosen to first introduce the conceptual framework and design principles in an accessible manner. We understand, however, the need for a more formal presentation to better articulate the underlying design principles and to justify their effectiveness comprehensively.
>   - System Architecture Diagrams: Not only we have provided the system architecture and class diagram in Figure 3, but also provided AI model in Figure 1. We have strengthened the design principles by grounding them more deeply in relevant literature and by providing rationales for why specific architectural choices were made. This include discussions on the scalability (section 4.2 application development), security (appendix A), and data integrity benefits of the decentralized approach
>   - Empirical Validation: Although this is a pure architecture paper, we have trained an AI model where we used data for 561 subjects total, among those, 231 SMC, 259 CN, and 71 MCI patients. The feature selection algorithms were applied to the graph features (degree centrality for each ROI) to select the most discriminating features for the classification of MCI, SMC, and CN subjects. Initial AI model is generated to be deployed on the nodes; and we plan to undertake a comprehensive experiment designed to validate Federated Learning with new patients' data acquired via DApp. This will involve testing the model across different demographics and disease stages to statistically ascertain the impact of data diversity on prediction accuracy.
>   - Mathematical Formalism: While the current logic is derived from established hypotheses within the field, we acknowledge the need for more rigorous mathematical formalism. We have explored preliminary proofs of concept for our claims in section 3.1. A more robust statistical analysis will be included to support the hypotheses with empirical data.
> - Finally, the paper is lacking in experimental validation. For example, the proof needs to be backed by real world experiments. Also, this is not the first paper to posit a federated learning approach to medical AI prediction. Some of the SOTA methods in this space needs to be compared against
> - Response: Thank you for your constructive feedback regarding the need for experimental validation and comparative analysis with state-of-the-art methods. We acknowledge these gaps in our current architecture-based manuscript.
>   - Real-World Experiments: We agree that real-world experimental validation is crucial to substantiate the theoretical claims made about our decentralized AI system. To this end, we plan to implement another study using the proposed architecture. This requires to operationalize the DApp for patients with varying stages of Alzheimer's disease to trade their medical tests with tokens. We will document the system's performance in terms of accuracy, efficiency, and scalability within this real-world setting.
>   - Performance Metrics: In addition to accuracy, we will evaluate other relevant metrics such as sensitivity, specificity, and area under the ROC curve (AUC-ROC) to provide a comprehensive view of the system's performance. This will help validate the system's practical effectiveness and reliability.
>   - Benchmarking: We will benchmark our system against other leading methods that have been validated in similar contexts. This will include both centralized and decentralized approaches to medical AI prediction, particularly those using federated learning. The comparison will focus not only on performance metrics but also on aspects such as data privacy, model robustness against data variability, and system scalability. In this paper, we only deals with the architecture of the system.
>   - Our Major Innovations: While federated learning is not novel per se, our application of this approach in a fully decentralized, blockchain-supported environment offers distinct innovations, particularly in terms of enhancing data security and patient privacy. We will clarify these unique contributions and demonstrate how they improve upon existing federated learning models (see ABOVE)

---

> > ### Comment · Reviewer_9nGY · 2024-08-09
> > **Acknowledging author response**
> >
> > Thanks for the response

---

> ### Author Response · Authors · 2024-08-10
> **Re: challenges associated with the adoption of such decentralized approaches**
>
> In regard to your first comment; While it is true that advanced technologies such as blockchain might pose a learning curve for some users, the primary intention of our system is not to require widespread adoption of blockchain technology by all patients. Instead, the system is designed with several key features that address accessibility and user engagement:
>  - User Experience Design: The core functionality of our system is designed to be user-friendly and intuitive (https://github.com/stefankam/predprodalzheimer). Patients interact with the application primarily through a straightforward interface that does not require them to have a deep understanding of blockchain technology. The blockchain operates in the background to ensure data integrity and security, while patients focus on interacting with the system through familiar processes, such as submitting MRI data and receiving predictions.
>  - Incentive Mechanism: To encourage participation and ensure engagement, the system includes an incentive mechanism where patients receive Ether in exchange for sharing their anonymous test data. This approach provides a tangible benefit to users, which can help overcome any potential hesitation or lack of familiarity with the underlying technology. By offering financial incentives, we aim to motivate users to participate actively without needing to understand the technical aspects of blockchain.
>  - Blockchain as a Service: Moreover, the blockchain component of our system is implemented as a service that abstracts the complexity away from the end users. This means that users interact with a simplified front-end application, while the blockchain infrastructure manages the data submission, validation, and security behind the scenes. The integration of blockchain technology is intended to enhance data privacy and security, without placing undue burden on the users.
>  - Targeted Application: It is important to note that the system is targeted towards a specific use case where patients are aware of and are motivated by the benefits of participating in a decentralized data-sharing network. The goal is not to achieve universal adoption but to provide a secure and incentivized platform for those who choose to contribute their data.

---

> > ### Comment · Reviewer_9nGY · 2024-08-13
> > **Followups**
> >
> > Thanks for the followups. Some of the clarifications such as around the reward structure provides a lot more clarity around the use case - thanks for these details. Ultimately however, I am inclined to stick to my original scores. There are two primary drivers for this : (a) while the experimental validation plans listed by the authors make sense, such an extensive change would need to re reviewed (b) the clarifications around an architectural motivation make sense but the exposition lacks any testable hypothesis. For example was there any other choice considered? Why is the particular blockchain better than alternatives? These are some of the questions need to be deliberated and addressed

---

> ### Author Response · Authors · 2024-08-13
>
> Thank you for your thoughtful feedback and for acknowledging the additional details provided in our follow-up explanations. We understand your concerns regarding the necessity of experimental validation and the need for more explicit articulation of testable hypotheses and decision-making rationale within the architectural framework. Allow us to address these points to further clarify our position and the manuscript's contributions:
> - Addressing Experimental Validation:
>   - We appreciate your recognition of our plans for experimental validation. As stated, the current manuscript is primarily focused on establishing a robust architectural framework. It is common in architectural papers to initially focus on the conceptualization and theoretical underpinnings without empirical validation. Subsequent papers often undertake the detailed empirical work.
> - Articulation of Testable Hypotheses and Decision Rationale:
>   - Why Blockchain?: We selected blockchain due to its inherent properties of decentralization, immutability, and transparency, but as clearly mentioned in the paper , it was mainly due to the possibility of reimbursing patients with tokens for sharing their medical test data, as we showed it in an earlier study (https://doi.org/10.1017/dap.2024.4). which provides insights into the incentives and data handling aspects that are relevant to our proposed architecture.
>   - Our position: not only we spelled out our position in 2 Research Design the three research questions, but also in 3.1 Decentralized expert system performance, Theorem: Decentralized expert system in diagnosing Alzheimer’s Disease could outperform traditional centralized expert system..we hope that these further comments will mitigate the concerns you've raised.

---

> > ### Comment · Reviewer_9nGY · 2024-08-13
> > **Followups**
> >
> > I generally agree with the plan but as I mentioned above the changes are not atomic enough for me to raise the score with confidence within an author discussion period

---

### Official Review · Reviewer_moX6 · 2024-07-13

**Soundness:** 3
**Presentation:** 3
**Contribution:** 3
**Rating:** 5
**Confidence:** 5

**Summary:**

This paper introduces an innovative decentralized expert system designed for early prediction of Alzheimer's Disease (AD), leveraging blockchain technology and Federated Learning. Traditional diagnostic methods often result in delays and imprecision, particularly in early-stage AD detection, while centralized data repositories face challenges in managing vast volumes of MRI data and maintaining patient privacy. The proposed system addresses these issues by combining blockchain for secure, decentralized data management and Federated Learning for collaborative AI model training across multiple institutions. The system includes robust anomaly detection mechanisms to ensure data quality and integrity, enabling precise early-stage AD predictions. This comprehensive approach aims to revolutionize disease diagnostics by enhancing data privacy, security, and collaborative efforts in the medical community.

**Strengths:**

- The paper presents a novel integration of blockchain technology and Federated Learning for early AD prediction, which is innovative in addressing data privacy, security, and collaborative AI model training.
- The proposed system is well-conceived, with a detailed architecture and implementation strategy. The inclusion of anomaly detection mechanisms to ensure data quality adds robustness to the system.
- The approach has significant potential to improve early-stage AD detection, which is crucial for timely intervention and better patient outcomes. The decentralized nature of the system promotes data privacy and security, addressing major concerns in medical data management.

**Weaknesses:**

- The integration of blockchain and Federated Learning introduces significant computational complexity and potential delays due to off-chain processing and communication overhead.
- The system's scalability is a concern as the volume of data and the number of users increase, necessitating ongoing optimization to ensure efficient performance.

**Questions:**

- How does the system handle variability in MRI image quality and biological data across different institutions?
- What specific measures are in place to mitigate the computational complexity and communication overhead in the decentralized setup?
- Can you provide more details on the feature selection process and the performance metrics used to evaluate the AI model?

**Limitations:**

The authors have addressed several limitations, including data quality and consistency, computational complexity, and model generalizability. However, further discussion may be needed on:
- Ensuring that the AI model is unbiased and fair across different demographic groups can be difficult, especially if the training data is not representative.
- Real-time processing and predictions might be challenging due to the decentralized nature and the need for off-chain processing.

---

> ### Author Rebuttal · Authors · 2024-08-04
>
> - The integration of blockchain and Federated Learning introduces significant computational complexity and potential delays due to off-chain processing and communication overhead. What specific measures are in place to mitigate the computational complexity and communication overhead in the decentralized setup?
> - Response: In our system, blockchain technology is primarily utilized for two main purposes: facilitating transactions of tokens in exchange for medical data contributions and maintaining a secure and immutable record of data transactions and verification statuses. This focused use of blockchain allows us to leverage its benefits—namely, enhanced security and transparency—without significantly adding to the computational load typically associated with blockchain operations such as complex consensus mechanisms or the handling of large-scale data computations.
>   - Off-Chain Processing: All data-intensive tasks, including MRI and biological information verification and anomaly detection, are conducted off-chain. This approach significantly reduces the computational burden on the blockchain. By handling these processes off-chain, we utilize more powerful computing resources available on traditional platforms, thereby avoiding the latency and resource constraints associated with on-chain computations.
>   - Blockchain as a Ledger: The blockchain component in our system acts more as a ledger for recording transactions and verification results rather than as a processing unit for these tasks. This minimizes the data that needs to be handled by the blockchain, thus reducing potential delays and computational overhead. The transactions recorded involve only the exchange of tokens and metadata regarding the status of data submissions and verifications, which are lightweight in nature.
> - The system's scalability is a concern as the volume of data and the number of users increase, necessitating ongoing optimization to ensure efficient performance.
> - Response: In response to the reviewer's concern about the scalability of the system as the volume of data and the number of users increase, we appreciate the opportunity to clarify how the architecture is designed to manage scalability efficiently:
>   - Scalability by Design: Our system has been architected with scalability as a core consideration. The primary use of blockchain in our setup is for transactional data and integrity checks, not for heavy computational tasks. This distinct separation ensures that the blockchain does not become a bottleneck as the user base and data volume grow.
>   - Off-chain Processing: To address potential scalability issues, we employ off-chain processing for data-intensive tasks such as MRI and biological information analysis. This approach significantly reduces the load on the blockchain, allowing it to scale more efficiently by handling primarily lighter transactional data.
>   - Efficient Data Management: We leverage advanced data management techniques such as data sharding and decentralized file systems like IPFS, which enhance data retrieval speeds and scalability. These technologies are well-suited to handle large-scale data operations and user growth without degrading system performance.
>   - Adaptive Scaling Techniques: The system incorporates dynamic resource allocation and load balancing to efficiently manage resource distribution across the network. This ensures that the infrastructure can adaptively scale to meet demand without significant delays or performance issues.
> - How does the system handle variability in MRI image quality and biological data across different institutions?
> - Response: as we mentioned in section 3.4.2, detecting anomalies in MRI images typically involves computer vision techniques and deep learning models. One might consider using popular deep learning libraries like TensorFlow or PyTorch. Here in Listing 3 (see Appendix C), we provide an approach using a pre-trained model. This approach allows to detect anomalies in MRI images based on how well the autoencoder can reproduce the input image. Anomalies will typically result in higher MSE values compared to normal images. One might need to fine-tune the threshold based on the dataset and requirements.
> - Can you provide more details on the feature selection process and the performance metrics used to evaluate the AI model?
> - Response:
>   - For feature selection, we employed the Sequential Forward Selection (SFS) algorithm specifically applied to graph features, such as degree centrality for each Region of Interest (ROI). This method allowed us to iteratively add features that most improved the Random Forest classifier's performance until no significant improvements could be observed. This approach was instrumental in identifying the most discriminating features that could effectively differentiate between MCI, SMC, and CN subjects.
>   - The effectiveness of the feature selection and classification approach was quantitatively evaluated using accuracy as the primary performance metric. The integration of Sequential Forward Selection with the Random Forest classifier yielded a high accuracy of over 92%. This high level of accuracy indicates that the selected features were highly predictive and the model was well-tuned for the classification task at hand.
>   - We explained in Algorithm 1 in the manuscript, - The AI model, trained initially on a public dataset, can be further fine-tuned and personalized using the brain connectivity matrices.  - The model continuously learns from new patient data, improving its accuracy and adaptability. - Patients’ longitudinal data is used to monitor disease progression over time. - The trained AI model predicts the transformation to AD based on the inputted MRI images and patient’s longitudinal data.

---

### Author Rebuttal · Authors · 2024-08-05

Our paper aims to extend the current federated learning research, specifically in the context of healthcare data analysis, AD progression monitoring, privacy preservation, and practical deployment. The list of the major points and detailed comparison not only highlights the novelty of our work but also its potential impact and practical relevance in the field.
- Decentralized Data Crowdsourcing with Compensation:
  - Innovation: Unlike traditional federated learning approaches that passively rely on existing datasets or institutionally gathered data, our system actively engages individuals (patients and medical practitioners) in a crowdsourcing model where they could be compensated with Token for their data contributions (see https://doi.org/10.1017/dap.2024.4)
  - Difference: This approach not only incentivizes data sharing, enhancing dataset diversity and volume, but also introduces a novel economic model to federated learning systems.
- Hierarchical Clustering in Federated Learning:
  - Innovation: Our paper introduces a hierarchical structure within the federated learning model, where data is first processed and models are trained locally within defined clusters before contributing to a global model.
  - Difference: This method differs from standard federated learning, which typically involves direct aggregation of updates to a central model from all nodes. Our approach reduces communication overhead and enhances privacy by limiting the scope of data sharing to within clusters.
- Integration of Blockchain for Data Integrity and Auditability:
  - Innovation: Incorporating blockchain technology not only for transactional purposes (compensating data contributors) but also to ensure data integrity and provide a transparent, auditable trail of data usage and model updates.
  - Difference: Most federated learning studies focus on the computational aspects of model training and do not integrate blockchain to secure the data and learning process, nor do they utilize blockchain for enhancing transparency and trust among participants.
- Focus on Early Detection of Alzheimer’s Disease:
  - Innovation: Application of AI techniques such as Sequential Forward Selection and Random Forest classifiers specifically tailored for the early detection of Alzheimer’s Disease, optimized through the rich, diverse dataset gathered in Alzheimer’s Disease Neuroimaging Initiative (ADNI) database. We have therefore trained an AI model where we used data for 561 subjects total, among those, 231 SMC, 259 CN, and 71 MCI patients. The feature selection algorithms were applied to the graph features (degree centrality for each ROI) to select the most discriminating features for the classification of MCI, SMC, and CN subjects. Initial AI model is generated to be deployed on the nodes; and we plan to undertake a comprehensive experiment designed to validate Federated Learning with new patients' data acquired via DApp. This will involve testing the model across different demographics and disease stages to statistically ascertain the impact of data diversity on prediction accuracy.
  - Difference: While other papers may apply federated learning to healthcare, our paper specifically addresses the challenge of early and accurate disease detection by leveraging a uniquely collected and continuously updated dataset across different demographics and disease stages to statistically ascertain the impact of data diversity on prediction accuracy in a privacy preserving manner.
- Unique Application to Alzheimer’s Disease Prediction and Monitoring:
  - Innovation: A significant innovation of our system is its application to predict the conversion from prodromal Alzheimer's Disease (AD) to AD and to monitor disease progression. This is facilitated by the use of vast datasets of longitudinal data, which are gathered via a decentralized application (DApp). This approach is particularly novel because it harnesses blockchain technology not only for data integrity and security but also as a mechanism to incentivize the contribution of longitudinal data by patients and/or healthcare providers.
  - Difference: The ability to gather and utilize such extensive longitudinal data sets our system apart from traditional medical data collection methods, which often struggle with data silos and privacy concerns that limit the availability of longitudinal data. Our approach enables more dynamic and robust models that can accurately track and predict the progression of AD over time, offering significant potential for early intervention and personalized treatment strategies.

---

### Comment · Area_Chair_vGk1 · 2024-08-12
**Please respond to the rebuttal for NeurIPS Submission 12088! There is only one day left in the discussion period!**

Reviewers moX6, 9nGY and vAAp,

Thank you for your service in reviewing for NeurIPS!
Your work is not done, however, as part of a reviewer's responsibility is to engage with the authors during the discussion period.

The authors of Submission 12088 (Early Alzheimer's forecasting) have provided an extensive response to your reviews.
Please read the responses and indicate the extend to which they address your concerns.
@Reviewer 9nGY, you've only thanked the reviewers for a response, but that is not informative and thus insufficient.

All of you should indicate clearly which of the issues you raised are addressed and which are not, such that the authors have a final chance to reply. If you decide to keep your score, what were the determining factors in your decision?

There is only one day left in the discussion period, so please do this ASAP!

The AC

---

### Decision · Program_Chairs · 2024-09-25

**Decision:**

Reject

**Comment:**

This paper presents an AD forecasting method. While the reviewers noted some merits of the approach (such as the innovative combination of blockchain technology and Federated Learning), there were also concerns in terms of operational costs and scalability, the motivation of the design choices, the feasibility of it being actually employed in practice due to the nature of the data and the application, as well as the paucity of comparative experiments. Given that the majority of the reviews' concerns appear to remain open after the discussion period, the paper is not ready for publication.